# Long-term disgust habituation with limited generalisation in care home workers

Charlotte Edgar[1], Faye Chan[1], Thomas Armstrong[2], Edwin S. Dalmaijer [1]*

1 School of Psychological Science, University of Bristol, Bristol, United Kingdom, 2 Department of Psychology, Whitman College, Walla Walla, Washington, United States of America

* edwin.dalmaijer@bristol.ac.uk

## Abstract

Countless workers handle bodily effluvia and body envelope violations every working day, and consequentially face deeply unpleasant levels of disgust. Understanding if and how they adapt can help inform policies to improve worker satisfaction and reduce staff turnover. So far, limited evidence exist that self-reported disgust is reduced (or lower to begin with) among those employed in high-disgust environments. However, it is unclear if this is due to demand effects or translates into real behavioural changes. Here, we tested healthcare assistants (N = 32) employed in UK care homes and a control sample (N = 50). We replicated reduced self-reported pathogen disgust sensitivity in healthcare workers compared to controls. We also found it negatively correlated with career duration, suggesting long-term habituation. Furthermore, we found that healthcare assistants showed no behavioural disgust avoidance on a web-based preferential looking task (equivalent to eye tracking). Surprisingly, this extended to disgust elicitors found outside care homes, suggesting generalisation of disgust habituation. While we found no difference between bodily effluvia (core disgust) and body envelope violations (gore disgust), generalisation did not extend to other domains: self-reported sexual and moral disgust sensitivity were not different between healthcare assistants and the control group, nor was there a correlation with career duration. In sum, our work confirms that people in high-frequency disgust employment are less sensitive to pathogen disgust. Crucially, we provide preliminary evidence that this is due to a process of long-term habituation with generalisation to disgust-elicitors within the pathogen domain, but not beyond it.

## Introduction

Disgust occurs in response to "offensive" stimuli [1] in order to avoid those associated with risk of contamination and illness [2]. It is counted among the basic emotions [3], and thought to be evolved in response to disease threat [4–7]. However, provided sufficient hygiene is observed, feelings of disgust could be unpleasant or counter-productive in certain contexts. Specifically, cleaning and medical professionals routinely encounter bodily effluvia (e.g. faeces) and the results of body envelope violations (e.g. wounds). This has the potential of causing significant discomfort through repeated disgust, and it remains largely unclear if and how individuals adapt to those potent disgust elicitors.

**Data Availability Statement:** All anonymised data and analysis scripts have been made available through Zenodo: https://doi.org/10.5281/zenodo.8325975.

**Competing interests:** The authors have declared that no competing interests exist.

In the short term, disgust appears difficult to overcome. It has been characterised as "cognitively impenetrable" [8] due to its resistance to approaches that aim to reduce negative emotions, such as cognitive restructuring [9] and extinction and counter-conditioning [10]. One reason could be that disgust is elicited by concrete rather than abstract properties of stimuli [8] and by simple associative links rather than complex appraisals [11]. Put more simply: you can't reason your way around a big stinky poo.

Non-cognitive mechanisms of (un)learning may also fail to reduce disgust. Habituation within a single experiment simply does not occur: oculomotor avoidance of a disgusting stimulus is equally strong from the first to the last trial, even if there were 24 trials of 12 seconds each (~5 minutes in total), and it remains unmoved after a period of encouraged direct exposure with a monetary incentive [12]. This is not the case for other negatively valenced stimuli, including fear-inducing and suicide-related stimuli, to which oculomotor responses readily habituate [12,13]. Sustained avoidance in this task is typically preceded by a brief (~1 second) period of initial approach to novel disgusting stimuli [12–14]. The task is therefore analogous to coming into contact with offensive stimuli in the wild: after perceiving a stimulus is offensive, the feeling of disgust drives avoidance of potential contaminants and is adaptive because it reduces the risk of illness.

Individuals may have more success habituation to disgust in the long term, but the existing evidence is limited. For example, cadaver work reduces medical students' disgust with body envelope violations over a period of several months, but does not generalise to other disgust domains or even to still-warm deceased bodies [15]. Disgust sensitivity is also lower among emergency physicians, but unrelated to the number of "disgusting" incidents [16]; suggesting a possible self-selection of less disgust-sensitive individuals into medicine. However, compared to medical students, qualified healthcare providers are less likely to disengage from patients due to disgust with their symptoms; which potentially suggests habituation through experience [17]. Further evidence comes from non-medical populations: disgust is subtly lower in mothers than childless women [18], and disgust for meat products is lower in butchers and deli workers compared to the general population [19].

The summarised studies mostly rely on self-report measures of disgust elicited by written or picture stimuli. Such self-report measures can have excellent psychometric properties, for example the internal consistency [6] and external agreement with others' ratings [20] of the Three-Domain Disgust Scale, or the high test-retest reliability of self-reported disgust for images [12]. While experimental tasks are often less reliable than self-report measures [21], behavioural disgust avoidance measured in preferential looking tasks has excellent internal consistency, Spearman-Brown $\rho$ = 0.81 to 0.94 and Cronbach's $\alpha$ = 0.78 to 0.91 [22], and test-retest reliability equal to self-reported disgust ratings [12]. Behavioural disgust avoidance correlates with self-reported disgust sensitivity, R = 0.28 to R = 0.34 [12], and with self-reported disgust ratings for stimuli, R = 0.19 to 0.50 [22]. However, a major difference between self-report and behavioural measures of disgust avoidance is that self-reported disgust shows a very large (Cohen's d = 1.25) placebo effect, whereas oculomotor avoidance (gaze dwell time) in preferential looking tasks only shows a small to medium (Cohen's d = 0.44) effect [23].

Self-reported disgust shows varied levels of habituation [12,24,25] that can occur even in the absence of behavioural changes [12]. Because of the difference in suggestibility between self-report and behavioural avoidance measures of disgust [23], it could be that the apparent difference in habituation between them reflects a demand effect: We think it likely that participants who are repeatedly asked to self-report disgust between being subjected to disgust-inducing stimuli are likely to understand researchers might expect them to reduce their ratings. We thus think that to better understand underlying emotional changes as a function of prolonged exposure to disgust elicitors, one must also measure behaviour.

In addition to being important for theory, understanding behaviour is also important in practice. Many workers, including cleaners and healthcare professionals, have to approach and handle potential contaminants. Better understanding whether and how their disgust adapts could help inform policy to improve job satisfaction and retention.

In the past, disgust avoidance behaviour has been quantified with eye-tracking measures. In preferential looking tasks, individuals look away from disgust elicitors and prefer to look at neutral stimuli instead. This pattern of sustained avoidance is unique to disgust, and does not occur for stimuli associated with anger, happiness, or suicide [12,13]. Eye tracking requires dedicated equipment, but can be replicated online using a blurred stimulus display with a cursor-locked aperture of high resolution. This method can be employed online, and has great psychometric properties: it has high reliability, correlates strongly with eye-tracking measures, and correlates with self-report to the same extent that eye-tracking measures do [22]. It is thus a highly practical way to measure disgust avoidance.

Here, we aim to answer two main questions: 1) Does disgust avoidance behaviour habituate over the course of long-term high-frequency exposure to bodily effluvia and body envelope violations, and 2) Does this habituation generalise to stimuli outside of this context? To do so, we study healthcare assistants in UK care homes and a non-medical control group. Based on outlined previous studies, we expect to see habituation to disgust elicitors within care home contexts, but no generalisation beyond those.

## Methods

### Participants

Two samples were recruited for the purpose of this study. One sample comprised healthcare assistants employed by non-NHS residential care homes in England. They were approached with permission from their employer and the School of Psychological Science Research Ethics Committee (approval code 10842), and their data was collected between 21 June and 2 August 2022. The other sample was a control group sampled from the general population (not healthcare workers), recruited through Prolific Academic with permission from the same ethics committee (approval code 11904), and their data was collected on 31 August and 1 September 2022.

The healthcare assistant sample was made up of 32 individuals; 21 of whom identified as female, 10 as male, and 1 preferred not to say; 20 identified as "White", 5 as "Asian or Asian British", 4 as "Black, Black British, Caribbean, or African", 2 as "mixed or multiple ethnic groups", and 1 as "other ethnic group". Their age was recorded in brackets, with 4 in the 18–21 years range, 10 in 22–25, 8 in 26–29, 5 in 30–33, 2 in 34–37, 1 in 38–41, none between 42–49, and 2 in 50+. Their career duration as a healthcare assistant in a care home ranged from 1 to 79 months, with a mean of 23.0, a median of 19.5, and a standard deviation of 19.9. Their work hours ranged from 15 to 40 hours per week, with a mean of 31.3, a median of 32, and a standard deviation of 7.8.

The control sample was recruited through Prolific Academic. We limited our recruitment to those residing (but not necessarily born) in the UK, to mirror the healthcare assistant sample. Out of a total of 50 participants, 25 identified as female and 25 as male; 39 identified as "White", 5 as "Asian", 2 as "Black", 2 as "mixed", and 2 were without ethnicity data. Their ages ranged from 21 to 65 years, with a mean of 35.28, median of 31.0, and standard deviation of 11.9. Divided into age brackets, there were 5 in the range of 18–21 years, 6 in 22–25, 10 in 26–29, 7 in 30–33, 5 in 34–37, 4 in 38–41, 5 in 42–49, and 8 in 50+. We were unable to exclude participants on the basis of fine-grained occupational data, so individuals with disgust-facing roles may have been included.

The samples were mostly well-matched, although the control sample had an equal male-female division, whereas the healthcare assistant sample skewed female. Previous sex differences have been reported in self-reported disgust sensitivity [26], but not in the behavioural disgust avoidance task employed here [12]. If such differences appeared in our results, it would have skewed the healthcare assistant group towards higher self-reported (primarily sexual) disgust sensitivity than the control group. (This was not the case.)

## Procedure

Healthcare assistants were recruited directly (in person and via email), and via word-of-mouth within their facilities. This was facilitated through author CE, who has experience as a healthcare assistant, and established contact with residential care homes. Upon agreeing to participate, they were directed to questionnaires hosted on Qualtrics, and from there to the behavioural disgust avoidance task hosted on Gorilla.

The control sample was recruited through Prolific Academic, from where they were directed to a study hosted on Gorilla that presented them with the behavioural disgust avoidance task and questionnaires. For both samples, written informed consent was obtained through our web-based experimental interface prior to participation.

Because participants completed the study from home, hardware limitations were less strict, and display size was not standardised. We allowed participation only through a computer (any operating system), using only Firefox of Google Chrome as browser (to ensure task compatibility).

For the healthcare assistant sample, we included specific questions on their professional experience. We asked how long they have been employed in care homes, how many hours they worked per typical week, and how often they encountered a range of specific disgust elicitors (e.g. soiled adult pads, faeces outside of the toilet, blood stains, or open wounds).

## Self-reported disgust sensitivity

We administered the Three-Domain Disgust Scale [26], which is comprised of 21 items equally divided between three subscales of pathogen, sexual, and moral disgust sensitivity. Questions were answered with a rating from 0 ("not at all disgusting") to 6 ("extremely disgusting"), and averaged within each subscale. This questionnaire was a product of thorough psychometric work, and as a result items load well onto their designated subscales, which have high internal consistency.

We also attempted to administer the Disgust Scale–revised [27,28], but failed to do so due to an error that caused only the first half of the questionnaire to be presented to the healthcare assistant sample.

## Behavioural disgust avoidance

Our behavioural disgust avoidance task was a preferential looking task, in each trial of which people are presented with two side-by-side presented images of a disgusting and a neutral content. Stimuli appeared in fixed positions to the left and right of the centre, but which stimulus appeared on which side was randomised. After a single practice trial with neutral stimuli, a 16 disgust-neutral stimulus pairs were repeated 4 times each. The order was randomised within each block of 16 unique stimulus presentations, and a break was offered halfway through the task. The presentation time of each stimulus was 10 seconds, the inter-stimulus interval was 1 second, and each trial started with a centrally-presented fixation marker that had to be clicked with the mouse cursor.

In eye-tracking versions of this task [12,13,29], eye movements are recorded with a specialised device. Because we were keen to accommodate healthcare assistants with testing-at-home,

we instead opted for a web-based version in which participants move a high-fidelity aperture through an otherwise blurred display to mimic natural vision. This approach has been validated to identify attentional biases for threatening [30] and sexual stimuli [31]; and has been shown to be reliable and valid in preferential looking tasks with disgusting or pleasant stimuli [22].

The task was hosted on the online experiment builder and presentation platform Gorilla [32], using its MouseView.js plug-in for stimulus presentation with a mouse-locked aperture [22]. This platform offers decent presentation and response timing accuracy [33], which were more than adequate considering our long presentation time of 10 seconds.

Stimulus images were sampled from the DIRTI stimulus set [34], and depicted faeces, mucous, sick (vomit), blood, and wounds; or neutral matches such as clean toilets, sinks, tissues, objects and environments, and unharmed body parts. The stimuli were subdivided into two equally sized categories for core (bodily effluvia) and gore (body envelope violations and other harms). The image pairs were then further subdivided into another two equally sized categories by a subject expert (author CE): those likely to be encountered within a care home, and those likely encountered outside of it.

The variable of interest was dwell time, or how long a participant hovered their mouse cursor over (and was thus able to observe) an image. This was recorded for both types of stimuli, as well as for any space outside of them.

## Data reduction and statistical analysis

For each individual, self-reported disgust sensitivity was computed as the average of all 7 questions within the pathogen, sexual, and moral subscales. These were then compared between the healthcare assistant and control groups using Welch's t-test, which is more reliable when samples have unequal size (as ours did). Bayes Factors for t-tests were computed using a Cauchy prior with scale factor 0.707 [35].

Within the healthcare assistant sample, self-reported disgust sensitivity scores were also correlated with the number of months participants had worked in care homes, using parametric ($R$) and non-parametric (Kendall's $\tau$) correlation coefficients. "Jeffreys exact" Bayes Factors [36] were also computed for correlations, using $\kappa = 1$.

Within each trial of the preferential looking task, dwell time was computed as the cumulative time that the mouse cursor hovered over each of three areas of interest: the disgusting stimulus, the neutral stimulus, and the display outside of those stimuli. These were than converted to proportions by dividing dwell time for each area by the trial duration.

We analysed dwell time proportions using linear mixed-effects analyses, with participant number as a random factor. We tested a number of models, ranging from highly complex (all factors included) to most basic (only testing for disgust-neutral differences). We did this for the combined samples, as well as for healthcare assistants and controls separately.

Dichotomous factors were group (healthcare assistant or control), stimulus (disgust or neutral), environment (stimulus from inside or outside care home context), disgust type (stimulus core or gore). Stimulus repetition (each stimulus was shown four times) was used as a continuous factor. Within each model, we also included all conceivable interaction effects between the included factors.

After running the linear mixed-effects analyses, we identified as best-fitting model that with the lowest Bayesian Information Criterion (BIC). We reported the difference in BIC between the best and second-best fitting models (ΔBIC). We interpreted these according to Raftery's guidelines, treating ΔBIC values over 10 as very strong evidence for the best-fitting model [37]. For readers' convenience, we also computed Bayes Factors (BF) from the BICs using the

approach outlined by Wagenmakers [38]. These quantified evidence in support of the best-fitting model compared to the second-best fitting model; and we interpreted them according to an adjusted interpretation [39] of Jeffreys' guidelines [40], which consider BF values of 1–3 "anecdotal", 3–10 as "moderate", 10–30 as "strong", 30–100 as "very strong", and over 100 as "extreme" evidence for the alternative hypothesis (e.g. difference or correlation); and the reciprocal values as evidence for the null hypothesis (e.g. no difference or no correlation).

The described analysis of preferential looking data is an established approach that has been applied in data from both eye-tracking [12,14] and MouseView.js [22]. Analyses were conducted in Python 3.8.10, using NumPy 1.22.0 [41], SciPy 1.7.1 [42], and Pingouin 0.5.3 [43]; and Matplotlib 3.4.3 [44] was used for data visualisation.

### Statistical power

The sample size of this study was primarily determined by practical limitations, such as access to healthcare assistants working in care homes. We thus provide power estimations for our achieved sample sizes (N = 32 healthcare assistants and N = 50 control) in Fig 1. Our statistical

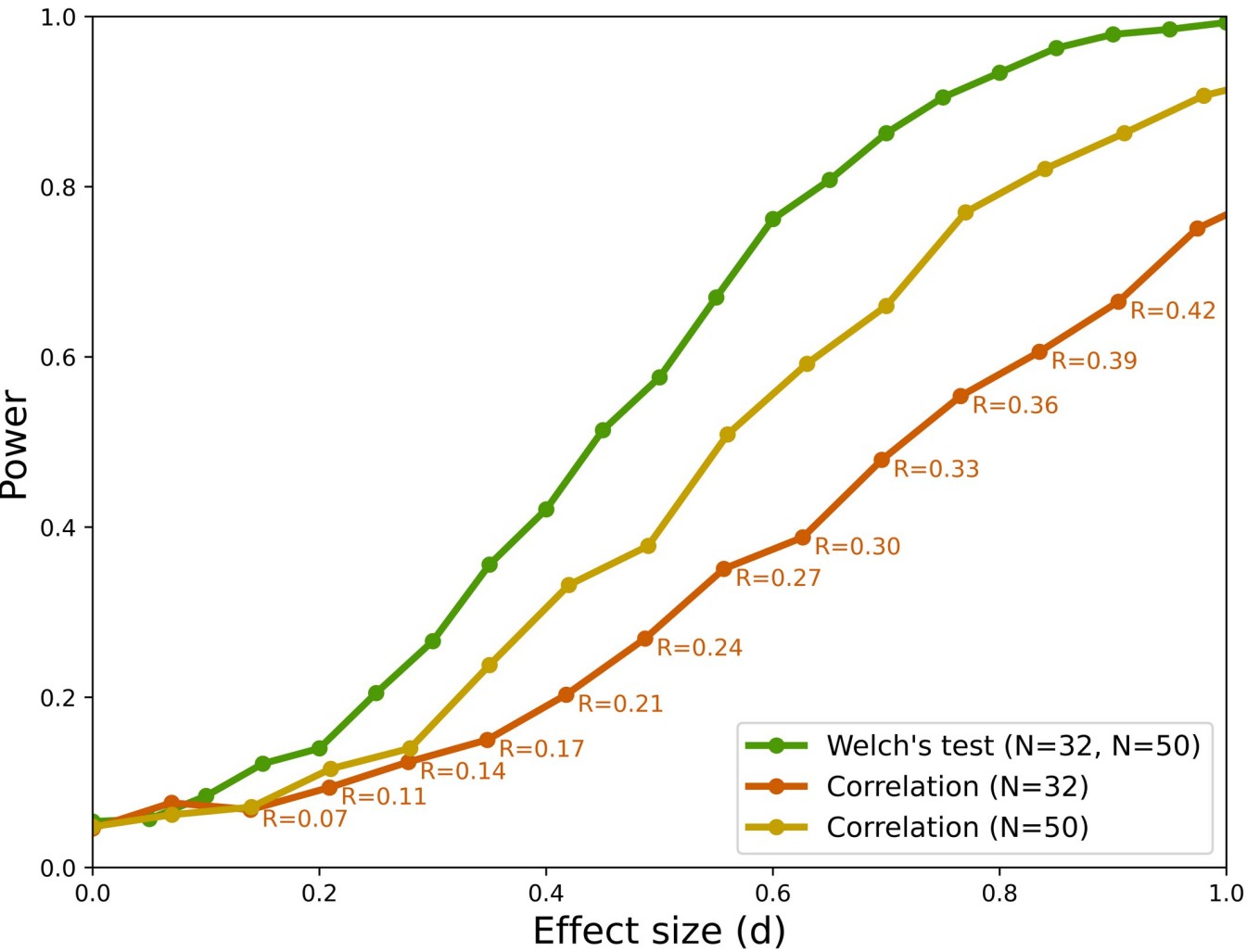

**Fig 1. Statistical power (y-axis) as a function of effect size (x-axis) for the sample sizes employed in the current study.** Power was computed as the proportion of simulated tests (N = 1000) that resulted in a statistically significant difference. To fit within the same plot, correlation coefficients R were converted to d, but for convenience R values are annotated.

power is low for direct tests of differences and correlations, which could result in erroneous rejection of the alternative hypothesis (Type II error). We have aimed to remedy this to some extent by contextualising *p* values with 95% confidence intervals on test statistics and with Bayes Factors to quantify the weight of evidence for the alternative or null hypotheses.

For linear mixed-effects analyses, power depends on the number of individuals and the number of observations per individual; and the recommended total number of observations per condition is 1600 [45]. In our study, the number of observations per condition is 5248 for the full sample (82 participants, 16 stimuli per condition, each with 4 repetitions), 2048 for the healthcare assistant sample, and 3200 for the control sample. Note that the cited recommendation is for research on response times, and aimed at "very small" effect sizes [45]. By contrast, the main effect of stimulus type (disgust vs. neutral) on dwell time in preferential looking tasks is rather large: β = 0.49 in a well-powered study with N = 101 and 4848 observations per condition [12]. In sum, while our sample size is limited, we made up for this with a relatively high number of observations per participant to meet recommendations for statistical power.

In addition to the above, we report Bayes Factors for comparisons between linear mixed-effects models, and 95% confidence intervals on effect estimates to contextualise *p* values.

## Results

### Manipulation check: Healthcare assistants frequently encounter disgust elicitors

We found that, as expected, healthcare assistants reported frequently encountering specific disgust elicitors in both the core and gore category. In the core disgust domain, less frequently encountered were stoma and catheter bags, and more frequent were encounters with (soiled) adult pads and faeces outside of the toilet. In the gore domain, blood stains and open wounds were relatively frequent. Frequency distributions for all queried elicitors are displayed in Fig 2.

### Disgust avoidance is lower in healthcare professionals than controls

For illustration purposes, we quantified disgust avoidance as the difference in dwell time proportion between the disgust and neutral stimuli, averaged across participants within each sample. The results are shown in Fig 3 (control group) and Fig 4 (healthcare assistants). Disgust avoidance was pronounced in the control group, but was absent in the healthcare assistants. Statistical analyses supported this observation, and are summarised below.

The best-fitting full model (ΔBIC = 78.57, BF = 1.15e17, "extreme" evidence for this model over the next-best) showed no main effects of group [β = 0.048, 95% CI (-0.145, 0.240), *t* = 0.49, *p* = 0.629], stimulus [β = -0.052, 95% CI (-0.106, 0.001), *t* = -1.91, *p* = 0.059], or repetition [β = 0.013, 95% CI (-0.025, 0.051), *t* = 0.68, *p* = 0.496]; nor an interaction between group and repetition [β = 0.032, 95% CI (-0.016, 0.081), *t* = 1.30, *p* = 0.197]. However, it did show interactions between group and stimulus [β = -0.435, 95% CI (-0.503, -0.366), *t* = -12.45, *p*<1e-16], suggesting that participants in the control group showed a larger difference in dwell time over disgusting and neutral images than healthcare assistants. There was also an interaction between stimulus and repetition [β = -0.167, 95% CI (-0.220, -0.113), *t* = -6.11, *p* = 3.311e-8], suggesting the difference between dwell time over disgusting and neutral stimuli increased with repeated presentations. Finally, there was an interaction between group, stimulus, and interaction [β = -0.131, 95% CI (-0.200, -0.063), *t* = -3.76, *p* = 3.16e-4], suggesting that the increase in dwell time difference between disgusting and neutral stimuli was larger for the control group compared to the care home group.

The best-fitting model's inclusion of the group factor (and its interaction with the stimulus factor) demonstrated that the control and healthcare samples were different in their disgust

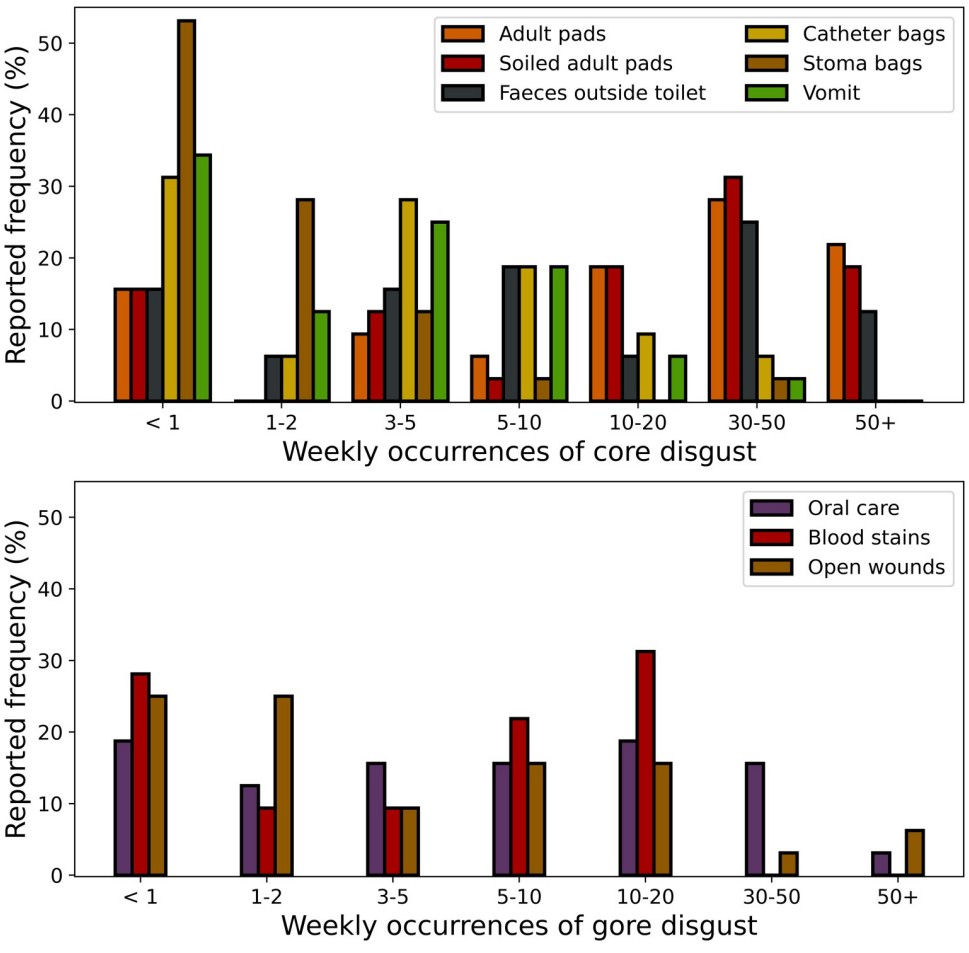

**Fig 2. Relative frequencies of reports from healthcare assistants of the number of weekly encounters with specific disgust elicitors.** The top plot depicts elicitors in the "core" category (mostly related to bodily effluvia), and the bottom plot the "gore" category (mostly related to body envelope violations).

avoidance. The equivalent model without the group factor fitted substantially worse than the best-fitting model (ΔBIC = 118.63, BF = 5.76e25, "extreme" evidence for this model over the next-best).

The lack of inclusion of factors environment (inside or outside care home context) and disgust type (core or gore) suggested that neither affected disgust avoidance. For the control group, this was likely because all subcategories were equally effective in inspiring disgust avoidance. For the healthcare assistant group, this was unexpected, and suggests that their lack of avoidance of disgusting stimuli generalised beyond the context of their care home.

Taken together, these findings revealed that the control sample showed more disgust avoidance than the healthcare assistants, and a larger increase in avoidance over repeated exposures after an initial subtle inclination towards the disgusting stimulus.

## Within-group confirmatory analyses

The linear-mixed effects analysis on both groups yielded interesting insights, but some might have been hard to interpret due to being two- and three-way interactions. The following linear-mixed effects analyses are conducted on each group separately to confirm that our interpretations hold.

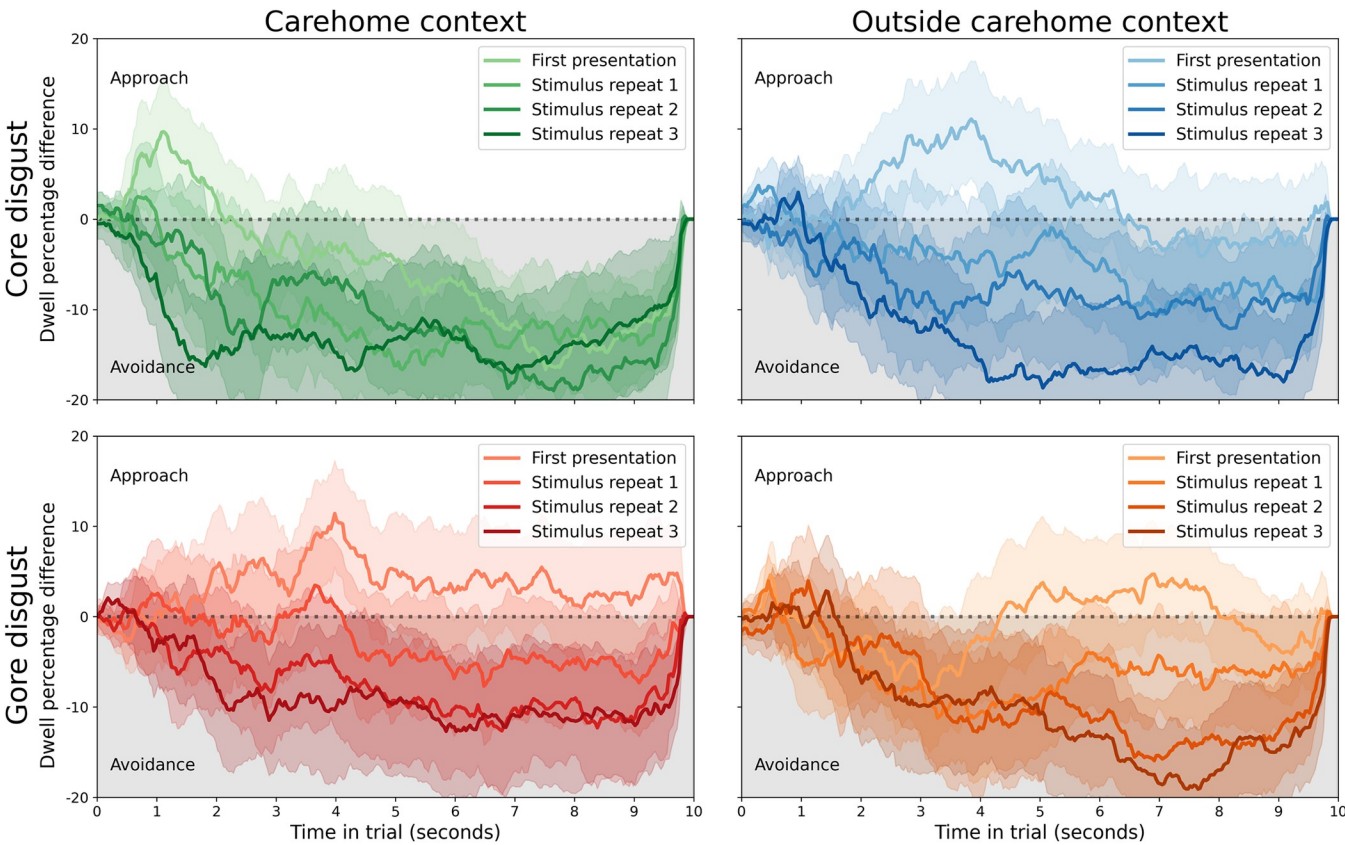

**Fig 3. Disgust avoidance in the control group.** Across conditions, participants showed stable disgust avoidance that increased with stimulus repetitions. Each line reflects the average difference in dwell time percentage between the disgusting and neutral stimuli, with shaded areas depicting the 95% confidence interval. The top row is for core disgust stimuli (images with bodily effluvia), the bottom for gore disgust (images related to body envelope violations). The left column is for stimuli that are often encountered within care homes, the right for stimuli encountered outside of that context.

Within the control group, the best fitting model ($\Delta$BIC = 33.65, BF = 2.03e7, "extreme" evidence for this model over the next-best) showed a main effect of stimulus [$\beta$ = -0.467, 95% CI (-0.509, -0.425), $t$ = -21.89, $p<$1e-16], a main effect of repetition [$\beta$ = 0.043, 95% CI (0.014, 0.073), $t$ = 2.88, $p$ = 0.006], and an interaction between stimulus and repetition [$\beta$ = -0.286, 95% CI (-0.328, -0.244), $t$ = -13.40, $p<$1e-16]. This confirms that the control group showed more dwell time for neutral compared to disgusting stimuli, and that this difference increased with repeated exposures. The lack of inclusion of factors environment (inside or outside care home context) and disgust type (core or gore) confirmed that neither affected disgust avoidance for the control group.

Within the care home group, the best fitting model ($\Delta$BIC = 38.85, BF = 2.73e8, "extreme" evidence for this model over the next-best) showed no main effect of stimulus [$\beta$ = -0.057, 95% CI (-0.113, -0.000), $t$ = -1.97, $p$ = 0.057], no main effect of repetition [$\beta$ = 0.014, 95% CI (-0.025, 0.054), $t$ = 0.71, $p$ = 0.486], and an interaction between stimulus and repetition [$\beta$ = -0.181, 95% CI (-0.238, -0.125), $t$ = -6.31, $p$ = 5.17e-7]. This suggests that while there were no substantial effects of either stimulus or repetition, healthcare assistants showed a subtle decrease (from non-significant approach to non-significant avoidance) over repeated stimulus presentations. The non-inclusion of factors environment (inside or outside care home context) and disgust type (core or gore) in the best fitting model confirmed that neither played a role in shaping disgust avoidance behaviour.

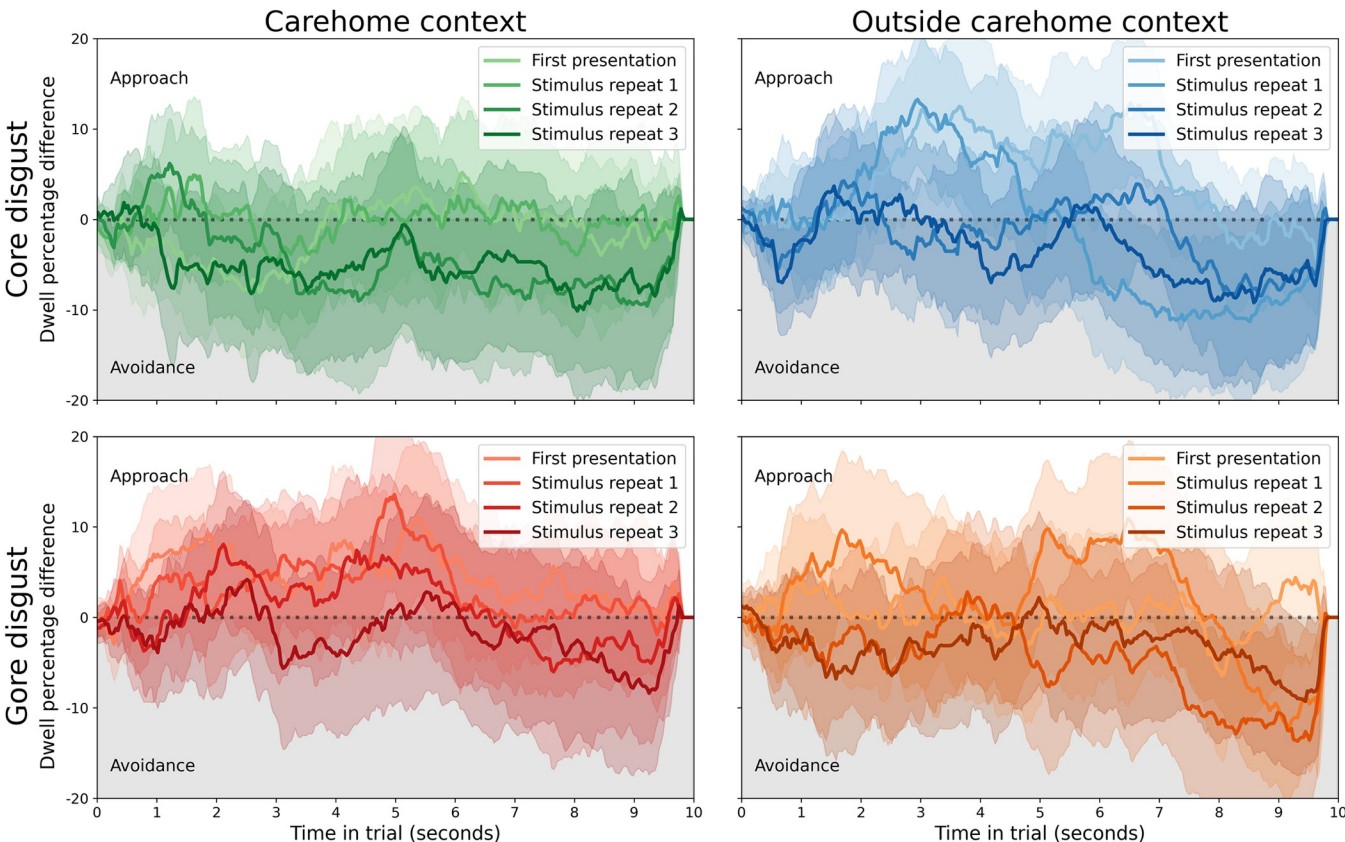

**Fig 4. An absence of disgust avoidance in the healthcare assistant group.** Across conditions, participants showed no bias towards either stimulus. Each line reflects the average difference in dwell time percentage between the disgusting and neutral stimuli, with shaded areas depicting the 95% confidence interval. The top row is for core disgust stimuli (images with bodily effluvia), the bottom for gore disgust (images related to body envelope violations). The left column is for stimuli that are often encountered within care homes, the right for stimuli encountered outside of that context.

## Self-reported pathogen disgust

When compared to scores from the original psychometric study of the Three-Domain Disgust Scale [26], our healthcare assistant sample scored substantially lower on all subscales [pathogen: $t(37.64)$ = -4.42, $p$ = 8.09e-5, BF10 = 1385, "extreme" evidence for a difference; sexual: $t(40.61)$ = -5.10, $p$ = 8.40e-6, BF10 = 25772, "extreme" evidence for a difference; moral: $t(33.21)$ = -3.23, $p$ = 0.003, BF10 = 21.8, "strong" evidence for a difference], and our control samples scored statistically significantly lower only on the sexual subscale [pathogen: $t(58.52)$ = -1.84, $p$ = 0.071, BF01 = 1.29, "anecdotal" evidence for no difference; sexual: $t(54.95)$ = -3.43, $p$ = 0.001, BF10 = 37.4, "very strong" evidence for a difference; moral: $t(56.13)$ = -1.59, $p$ = 0.118, BF01 = 1.93, "anecdotal" evidence for no difference]. This could reflect differences in cultural background (original in US, ours in UK), period during which data was collected (original in 2008 or before, ours in 2022), and participant age, maturity, and socioeconomic background (original sample from university undergraduates, ours from individuals from a higher age range and ~80% non-student).

More importantly, when compared directly, self-reported pathogen disgust sensitivity was lower for healthcare assistants [$M$ = 3.08, $SD$ = 0.95] than controls [$M$ = 3.57, $SD$ = 1.09], $t(72.50)$ = -2.09, CI 95% (-0.95, -0.02), $p$ = 0.040, BF10 = 1.52, "anecdotal" evidence for a difference. There was no difference in self-reported sexual disgust sensitivity between healthcare assistants [$M$ = 2.37, $SD$ = 0.99] and controls [$M$ = 2.61, $SD$ = 1.39], $t(78.95)$ = -0.90, CI 95%

(-0.77, 0.29), $p$ = 0.370, BF01 = 2.99, "anecdotal"/"moderate" evidence for no difference; or in self-reported moral disgust sensitivity for healthcare assistants [$M$ = 2.98, $SD$ = 1.22] and controls [$M$ = 3.43, $SD$ = 1.15], $t$(63.11) = -1.65, CI 95% (-1.00, -0.1), $p$ = 0.104, BF01 = 1.32, "anecdotal" evidence for no difference. This suggested that healthcare assistants could be less sensitive to pathogen disgust than the control group, but that statistical evidence for this was weak and ripe for replication in a larger sample. The potential difference could be a result of habituation in healthcare assistants, but it could also be a pre-existing difference.

There was a statistically significant correlation between the number of months healthcare assistants worked in a care home [M = 23.03, SD = 19.58] and their self-reported pathogen disgust sensitivity [$\tau$ = -0.28, $p$ = 0.027; $R$ = -0.38, CI 95% (-0.64, -0.04), $p$ = 0.031, BF10 = 2.03, "anecdotal" evidence for correlation]; but not for sexual [$\tau$ = -0.14, $p$ = 0.274; $R$ = -0.19, CI 95% (-0.50, 0.17), $p$ = 0.303, BF01 = 2.73, "anecdotal" evidence for no correlation] or moral disgust sensitivity [$\tau$ = -0.22, $p$ = 0.093; $R$ = -0.23, CI 95% (-0.54, 0.13), $p$ = 0.205, BF01 = 2.11, "anecdotal" evidence for no correlation]. This suggested that healthcare assistants' pathogen disgust sensitivity reduced over the duration of their careers, or that more disgust-sensitive healthcare assistants were more likely to exit the profession. Note that the statistical evidence was weak, and would ideally be replicated in a larger sample in future work.

A reduction over the duration of healthcare assistants' careers was not evident in average behavioural disgust avoidance [$\tau$ = -0.04, $p$ = 0.757; $R$ = 0.07, CI 95% (-0.29, 0.41), $p$ = 0.718, BF01 = 4.27, "moderate" evidence for no correlation]. However, this could well be due to a floor effect, as this sample's mean avoidance was indistinguishable from 0 [$t(32)$ = -0.34, CI 95% (-0.07, 0.09), $p$ = 0.736, BF01 = 5.03, "moderate" evidence for no difference].

## Discussion

Healthcare assistants working in care homes did not show behavioural disgust avoidance, where a non-medical control group did. This was true for stimuli that depicted bodily effluvia and (the results of) body envelope violations, i.e. core and gore disgust. Unexpectedly, healthcare assistants also lacked behavioural avoidance of stimuli that typically occur outside of care homes, whereas the control group showed typical avoidance. This suggested that care home workers habituated to the types of disgust that they face on a daily basis, and that this habituation generalised to all core and gore disgust elicitors.

We also found that healthcare assistants self-reported lower pathogen disgust sensitivity compared to controls, which could reflect habituation among care home workers or a pre-existing difference between the groups. Self-reported pathogen disgust sensitivity was negatively correlated with how long healthcare assistants were employed in care homes, suggesting it reduced over prolonged and repeated exposure to disgust elicitors, consistent with a long-term habituation account. This pattern of results did not hold for sexual or moral disgust sensitivity, suggesting no generalisation of disgust habituation to different domains.

### Evidence for disgust habituation

Our results are concordant with previous evidence of reduced self-reported disgust as a function of exposure in medical [15–17] and non-medical environments [18,19]. They also corroborate the previous study of time-in-job and reduction in self-reported disgust for meat products in butchers and deli workers [19].

Crucially, our findings support evidence of habituation in self-reported disgust sensitivity with corroborating behavioural data. This is an important piece of triangulation, because self-report measures and actual avoidance behaviour do not always align in disgust [12,24,25]. For example, minor reductions in self-reported disgust (d = 0.38 in two separate experiments)

have been paired with a statistically significant absence of habituation in oculomotor disgust avoidance in a short-term experimental study [12], In the introduction, we suggested this could be due to demand effects. However, we think our findings in healthcare assistants suggest that long-term habituation to disgust occurs in both self-report and behavioural avoidance.

## Evidence for limited generalisation of habituation

To our surprise, healthcare assistants did not only show a lack of disgust avoidance for stimuli that can be found in care homes, but also to those outside of this context. This is evidence for generalisation of habituation from the environment in which it occurred to elicitors from outside this environment. It was a surprising finding, as previous reports suggested highly limited generalisation in medical students whose work on cadavers habituated them to cold but not warm bodies [15]. This difference could be due to response modality: our results are behavioural, whereas previous evidence comes from self-report. However, it could also be an effect of experience: our sample of healthcare assistants had worked in care homes for an average of almost two years, whereas the students in the previous study had only been enrolled a single first-year gross anatomy class at university that spanned 2–3 months. Perhaps disgust only generalises on the larger timescale of exposure in the present study.

In self-report measures, both core and gore disgust elicitors load onto the same "pathogen disgust" factor [26]. However, this could be due to a lack of number and richness of items in the employed questionnaire, which does not include a great deal of body envelope violations. Indeed, previous studies have shown differences in response profiles between core and gore disgust. Bodily effluvia inspire a physiological change in stomach rhythm whereas body envelope violations alter heart-rate variability [46]. In addition, the two share a different but partially overlapping response profile in oxygenated blood in the brain [47]; and in alpha power in the electroencephalogram, although this could simply reflect differences in arousal [48]. It has been argued that this difference is due to vicarious feelings inspired by empathic simulation of observed body envelope violations, which is then verbalised as "disgust" despite it being qualitatively different to disgust to bodily effluvia [49].

Despite the above, we observed no difference in avoidance between the two stimulus categories in the control or the healthcare assistant group. This is not necessarily a limitation of the method, as differences in behavioural avoidance between subcategories of disgust elicitors (bodily effluvia versus spoiled food) have previously been show [13]. Healthcare assistants' lack of behavioural avoidance of both bodily effluvia and bodily harm could thus reflect a genuine habituation to each separate category.

In sum, long-term habituation of disgust avoidance generalised to stimuli found outside the set of habituated stimuli within the overarching "pathogen disgust" category. However, reductions in self-reported "sexual disgust" and "moral disgust" sensitivity were not found. We interpret this pattern as limited generalisation of disgust habituation within the pathogen domain, but not beyond it.

## Potential implications for healthcare staff retention

This is a small study, and should thus only be taken as preliminary. However, the notion of long-term habituation to disgusting stimuli aligns with anecdotal reports from those who come into close contact with them on a regular basis, like parents and healthcare professionals who report being less bothered by soiled nappies and pads over time. Because the high levels of disgust are profoundly unpleasant, they are likely to negatively impact staff retention, which is a problem for 48% of care homes in the UK [50]. Thus, reducing disgust in healthcare professionals could help to combat workforce challenges that negatively impact the sector.

Within the lab, such habituation does not occur in the short term, not even after encouraged exposure through monetary incentives [12]. However, previous efforts were successful in reducing disgust avoidance with a combination of encouraged exposure and domperidone, a peripheral dopamine antagonist that normalises gastric rhythm and barely crosses the blood-brain barrier [14].

Our current findings suggest that long-term disgust habituation may occur in healthcare careers. We think this habituation could potentially be encouraged with pharmacological support in the early stages of employment, thereby leading to professionals developing earlier reductions in disgust avoidance and greater job satisfaction. However, further research is warranted before implementation of this type of measure: while disgust is unpleasant, it is also adaptive. The effect of a single dose of domperidone on disgust avoidance was subtle, but it is unknown if sustained usage could push individuals towards maladaptive behaviour such as reduced hand-washing.

## Limitations

An obvious limitation of this study is the relatively small sample size of N = 32 in the healthcare assistants sample and N = 50 in the control sample. This was due to practical limitations on access to care home facilities and their staff, and we aimed to remedy this by including a relatively large number of trials per condition in the preferential looking task (64 trials per condition per participant). In addition, where this was not possible, we have aimed to report statistics in such a way that the uncertainty in estimates is clearly reflected in both confidence intervals and Bayes Factors.

One alternative explanation for the differences between healthcare assistants and the control group is selection bias. Disgust sensitivity partially predicts enrolment into different specialities and courses among medical students, with those scoring lower being more likely to choose more hands-on careers like nursing and surgery [51–53]. We cannot rule out that our sample of healthcare assistants started started their careers with lower disgust sensitivity. However, the correlation between disgust sensitivity and time worked in healthcare suggests that habituation also occurred on top of any pre-existing difference.

It is harder to rule out longitudinal selection bias. Specifically, it could be that individuals with lower disgust sensitivity have a lower yearly probability of quitting. This would mean that, within a sample of healthcare professionals, those with lower disgust sensitivity are more likely to be overrepresented among individuals with longer careers. This could result in an artificial correlation between career duration and disgust sensitivity. The only way to fully exclude this alternative interpretation is with a longitudinal design and a survival analysis.

## Conclusion

Self-reported pathogen (but not sexual or moral) disgust sensitivity is lower in healthcare assistants who work in care homes compared to a control sample, and correlates with the duration of employment. In addition, behavioural avoidance of core (bodily effluvia) and gore (body envelope violations) disgust stimuli found inside and outside care home contexts was absent in healthcare assistants, but present in the control group. This suggests prolonged and frequent exposure to contaminants (faeces, sick, mucous, blood, wounds, etc.) induces habituation that generalises to non-healthcare stimuli, but not outside of the pathogen disgust domain.

## Acknowledgments

CE and FC were students on the *MSc in Psychology (Conversion)* course at the University of Bristol.

## Author Contributions

**Conceptualization:** Charlotte Edgar, Edwin S. Dalmaijer.

**Data curation:** Charlotte Edgar, Edwin S. Dalmaijer.

**Formal analysis:** Edwin S. Dalmaijer.

**Funding acquisition:** Edwin S. Dalmaijer.

**Investigation:** Charlotte Edgar, Faye Chan.

**Methodology:** Charlotte Edgar, Faye Chan, Thomas Armstrong, Edwin S. Dalmaijer.

**Software:** Edwin S. Dalmaijer.

**Supervision:** Edwin S. Dalmaijer.

**Visualization:** Edwin S. Dalmaijer.

**Writing – original draft:** Edwin S. Dalmaijer.

**Writing – review & editing:** Charlotte Edgar, Faye Chan, Thomas Armstrong, Edwin S. Dalmaijer.

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
