## [Decision Letter · Decision Letter 0]

29 Nov 2023

PONE-D-23-30077Long-term disgust habituation with limited generalisation in care home workersPLOS ONE

Dear Dr. Dalmaijer,

Thank you for submitting your manuscript to PLOS ONE. After careful consideration, we feel that it has merit but does not fully meet the expectations for a publication decision at this point. Therefore, we invite you to submit a revised version of the manuscript that addresses the points raised during the review process. In your revision, please pay careful attention to the details as raised by both reviews. You will need to address a few important issues, particularly regarding the statistical power and the coverage of the related literature. It is equally important to clarify the experimental concerns raised by R2. I also recommend that you associate the subject of disgust habituation with the ongoing research on behavioral disgust avoidance in political science. That may broaden the contributions of your work and shed light on how the public is different from health care workers on the subject of disgust habituation.

We look forward to receiving your revised manuscript.

Kind regards,

Cengiz Erisen

Academic Editor

PLOS ONE

Journal Requirements:

Reviewers' comments:

Reviewer's Responses to Questions

**Comments to the Author**

1. Is the manuscript technically sound, and do the data support the conclusions?

Reviewer #1: Yes

Reviewer #2: Partly

2. Has the statistical analysis been performed appropriately and rigorously? 

Reviewer #1: Yes

Reviewer #2: Yes

3. Have the authors made all data underlying the findings in their manuscript fully available?

Reviewer #1: Yes

Reviewer #2: Yes

4. Is the manuscript presented in an intelligible fashion and written in standard English?

Reviewer #1: Yes

Reviewer #2: Yes

5. Review Comments to the Author

Reviewer #1: This manuscript is well-written, technically sound, and relevant to the research area it would be housed in. It does have a few rough edges that could be smoothed in revision.

ISSUE 1: STATISTICAL POWER

The manuscript reports multiple null findings (differences between "core" and "gore" stimuli; differences between home and work environment), but it reports no power analysis, nor does it comment on statistical power. Some reflection on likelihoods of Type II errors are needed. Further, better descriptions of point estimates and confidence intervals would help readers understand the precision of the estimates and how high or low group/context/stimulus differences might be.

ISSUE 2: INTERPRETATION OF HABITUATION EFFECT

The manuscript detects a negative relation between disgust sensitivity and length of time working in health care facilities. It interprets this effect as evidence of habituation. An alternative is (at least) equally plausible: that disgust sensitivity negatively predicts exiting the field. The manuscript mentions this possibility only in the discussion. It should be mentioned more prominently earlier in the manuscript, as should the possibility of selection bias (explaining differences between health care and control groups). The design can't distinguish between these explanations for the data, and favoring one (in my mind equally-plausible) explanation except at the very end of the manuscript risks misleading readers.

ISSUE 3: SOME RELEVANT LITERATURE MISSED

The manuscript would benefit from citation/discussion of at least two relevant papers.

First, Kupfer (2018) presents a compelling case that at least some of the disgust reported toward injuries is distinct from pathogen disgust. The manuscript makes a good point that assessments of "animal reminder" disgust are nearly indistinct from "core" disgust (citing Tybur et al., 2009). That (minimal) distinction might be limited to the items on the DS-R rather than pictures of dislocated joints and broken bones, though. Some commentary on this issue would be nice.

Second, Karinen et al. (2023) reports high self-other agreement in (pathogen) disgust sensitivity. The manuscript currently states, "This is an important piece of triangulation, because self-report measures and actual behavior do not always align in disgust." This sentence should be more precise (what does "always" mean, what categorizes a measures as "actual" behavior, what is the validity of such assessments, and what type of effect size does "align" refer to?). It should also direct readers to evidence for the validity of disgust sensitivity instruments, such as that reported by Karinen et al.

A few other comments:

* "Disgust occurs in response to offensive stimuli." This description is circular (see Tybur et al., 2013). Stimuli are considered "offensive" because they elicit disgust. This type of definition effectively boils down to "Disgust is elicited by disgusting stimuli."

* "A major risk here is that disgust-exposed participants simply displayed demand effects..." There's no evidence for this possibility, and the manuscript doesn't describe any reason to suggest that the possibility is plausible (or worth considering).

* "...one must thus measure behavior." There's a widespread assumption that behavioral measures are more valid than self-report ones. Dang et al. (2020) are (to me, persuasively) that behavioral measures are often LESS VALID than self-report ones because of their poor reliability and ambiguous validity (relative to many self-report measures). I'm not arguing that the behavioral measures adopted here suffer from these limitations. But default skepticism toward self-report measures in favor of "behavioral" ones is misguided.

* The manuscript describes Kendall's tau as more powerful than Pearson or Spearman r. I think that this is wrong, based on my own background in nonparametric statistics and on some sources I found online (https://www.ncss.com/wp-content/themes/ncss/pdf/Procedures/PASS/Power_Comparison_of_Correlation_Tests-Simulation.pdf). I looked at the manuscript cited in support of Kendall's being more powerful, and I don't think that it actually supports that statement. There ARE reasons to favor Kendall's over the others, but I don't think power is such a reason. Regardless, I urge the authors to carefully review this issue.

* In the "within group confirmatory analyses" section, the term "and an interaction" appears twice. Please clarify - interaction between which variables?

That's it from me. Hope the comments are helpful. And I commend the authors on writing a fine manuscript and asking an interesting research question.

Reviewer #2: Review of PONE-D-23-30077: Long-term disgust habituation with limited generalisation in care home workers.

The manuscript reports a study that compares two groups of participants—healthcare assistants working in residential care homes and a control group. Participants do a task in which it is measured how long they look at pictures that are disgusting versus not-disgusting. Participants also self-report their disgust sensitivity. The results show that participants in the control group look more at the not-disgusting than disgusting images. This difference is not observed for the healthcare assistants. Results also show that self-reported pathogen disgust sensitivity is lower for the healthcare assistants and that it correlates with the time they are working as healthcare assistants (those having worked longer reporting lower disgust sensitivity).

Overall, I like the research reported in this manuscript, but I think some things should be improved before publication.

Positive remarks:

(1) I think the research question is important. If and how there is habituation for disgust is a puzzle. Over a decade ago, I wrote a grant proposal on this question. It did not get funded. On the one hand, habituation occurs for many stimuli, suggesting that there should be habituation for disgust-elicitors. On the other hand, a straightforward hypothesis of a functional perspective is that there should be no habituation. If the purpose of a disgust response is to reduce contact with contagious objects—then in order to fulfil that function, there should be no habituation. (Or at least, it is difficult to see how disgust could serve the function of avoiding contamination if it did habituate after repeated exposure to contaminants.)

(2) I agree with the assessment of the authors that there is currently little empirical work on the habituation of disgust. Hence, all rigorous empirical work is welcome in my opinion.

(3) I appreciate the clear mention of the limitations regarding selection bias and longitudinal selection bias. While this certainly is a limitation, I think it is better to report the results (descriptively and with acknowledging this limitation) rather than to block publication for doubts about causal inference.

Suggestions for improvement:

(4) In the section “Participants”, please clarify if the participants in the control condition were/were not working as healthcare assistants. The current formulation is ambiguous. If it is possible that the participants in the control group working as healthcare assistants, then I think this has substantial consequences for interpreting the results. Then the control group would no longer be a control group.

(5) In my view, this work needs a clear discussion of statistical power. The groups were relatively small (N = 32 for the group of healthcare assistants, N = 50 for control). Please report sensitivity (i.e., what effect size could likely have been detected with this sample size), for the behavioral avoidance task, but also for the results for self-reported disgust sensitivity (for comparisons between groups as well as the correlations with career time).

(6) In the introduction, it says “In the past, disgust avoidance behaviour has been quantified with eye-tracking measures. In preferential looking tasks, individuals look away from disgust elicitors and prefer to look at neutral stimuli instead.” Could the authors elaborate how this relates to the assumed function of pathogen disgust? They write that “you can’t reason you way around a big stinky poo.” Yes, but to avoid touching a poo, it also does not help to look away. To avoid stepping in dog poo, you must look where it is and step aside. In this context, please also clarify what is meant with adaptive/maladaptive (alluded to in the first paragraph of the introduction). I can see how in an experimental task, it is “adaptive” to look the other way (avoiding unpleasant feeling), and that for healthcare workers it might appear “maladaptive” to feel so much disgust during their work. However, this usage of adaptive and maladaptive are different from the meaning of these terms in a functional evolutionary perspective: Disgust is likely the output of some adaptation—probably an adaptation for pathogen avoidance, so one could say that it is adaptive to feel disgust towards pathogen hazards, but maladaptive to feel disgust towards objects that are safe.

(7) The authors mention as a possible practical implications that job satisfaction among healthcare workers might be increased with pharmacological support that facilitates habituation of disgust responses. I am not sure this is a good idea. In my view, it would be good to mention the possibility that this has perverse (adverse) consequences. For example, suppose that some drugs makes these healthcare workers have such dampened disgust responses that they do not wash their hands after touching a big stinky poo. Or start having fun by throwing poo at each other. Is that “adaptive”?

6. PLOS authors have the option to publish the peer review history of their article (what does this mean?). If published, this will include your full peer review and any attached files.

Reviewer #1: No

Reviewer #2: No

---

## [Author Response · Author response to Decision Letter 0]

5 Jan 2024

Please note that the following information is also uploaded in a more readable PDF version, and that a document with tracked changes is also attached.

5 January 2023,

Re. PONE-D-23-30077 - Long-term disgust habituation with limited generalisation in care home workers

Dear Dr Erisen and esteemed reviewers,

Many thanks for considering our manuscript, and for providing constructive feedback. Please find enclosed a point-wise response to highlight the changes we have made on the basis of your reviews, and a copy of our revised manuscript in which additions are highlighted.

Yours sincerely,

Dr Edwin Dalmaijer

EDITOR:

In your revision, please pay careful attention to the details as raised by both reviews. You will need to address a few important issues, particularly regarding the statistical power and the coverage of the related literature.

ANSWER: We have included a new section on statistical power, which includes a sensitivity analysis through simulation. In addition, we have included further detail in our Results section to quantify confidence in estimates and evidence. We think our efforts comprehensively addressed both reviewers’ concerns about statistical power.

We have also improved the coverage of the literature following the excellent recommendations by Reviewer 1, and improved our discussion of questionnaire versus task-based measures in our study following important points raised by Reviewer 1. Furthermore, we addressed Reviewer 1’s suggestions around habituation, and their additional points.

It is equally important to clarify the experimental concerns raised by R2. 

ANSWER: In addition the the improvements in addressing statistical power requested by both reviewers, we have added detail on sampling, and we have improved our discussion of the (mal)adaptive nature of disgust in response to Reviewer 2’s suggestions.

I also recommend that you associate the subject of disgust habituation with the ongoing research on behavioral disgust avoidance in political science. That may broaden the contributions of your work and shed light on how the public is different from health care workers on the subject of disgust habituation.

ANSWER: We have deliberately kept our manuscript narrowly focussed on disgust to bodily effluvia and body envelope violations, as this pertained to our hypotheses, participants, and stimuli. Tying pathogen disgust to sociomoral and political opinions requires precise study designs, and is considered a bit of a “white whale” in disgust research (Strohminger, 2014, Philosophy Compass). As the Editor is aware, how disgust impacts political views is an active specialist subfield, and we would like to avoid treading into it without an a priori design aimed at doing so.

We do note in our manuscript that moral disgust sensitivity were not different between control participants and care home workers, and hope that this and the above satisfies the Editor’s curiosity:

“There was no difference in self-reported sexual disgust sensitivity between healthcare assistants [M=2.37, SD=0.99] and controls [M=2.61, SD=1.39], t(78.95)=-0.90, CI 95% (-0.77, 0.29), p=0.370, BF01=2.99, “anecdotal”/”moderate” evidence for no difference; or in self-reported moral disgust sensitivity for healthcare assistants [M=2.98, SD=1.22] and controls [M=3.43, SD=1.15], t(63.11)=-1.65, CI 95% (-1.00, -0.1), p=0.104, BF01=1.32, “anecdotal” evidence for no difference.” (section: “Self-reported pathogen disgust”)

Reviewer #1

This manuscript is well-written, technically sound, and relevant to the research area it would be housed in. It does have a few rough edges that could be smoothed in revision.

ANSWER: We thank the reviewer for their positive appraisal, and for their constructive review.

ISSUE 1: STATISTICAL POWER

The manuscript reports multiple null findings (differences between "core" and "gore" stimuli; differences between home and work environment), but it reports no power analysis, nor does it comment on statistical power. Some reflection on likelihoods of Type II errors are needed. Further, better descriptions of point estimates and confidence intervals would help readers understand the precision of the estimates and how high or low group/context/stimulus differences might be.

ANSWER: This is a very important point, and a new section on statistical power is now included in the methods section. In addition, 95% confidence intervals are now included for test statistics in the result section, to contextualise yes/no decisions of statistical significance on the basis of p-values. We have also included Bayes Factors to quantify the evidence in favour of alternative of null hypothesis (for tests of differences or correlation) or in favour of a specific model compared to others (for linear mixed-effects analyses).

Our sample sizes are indeed relatively small, and this is a concern for tests of differences and correlations of averages (e.g. mean dwell time within a single condition) and/or single values (e.g. duration of work in a care home or TDDS subscale score). This is now discussed in the Methods section, and power for the exact sample sizes is now shown in the newly added Figure 1:

“The sample size of this study was primarily determined by practical limitations, such as access to healthcare assistants working in care homes. We thus provide power estimations for our achieved sample sizes (N=32 healthcare assistants and N=50 control) in Figure 1. Our statistical power is low for direct tests of differences and correlations, which could result in erroneous rejection of the alternative hypothesis (Type II error). We have aimed to remedy this to some extent by contextualising p values with 95% confidence intervals on test statistics and with Bayes Factors to quantify the weight of evidence for the alternative or null hypotheses.” (section “Statistical power”)

Furthermore, 95% confidence intervals on test statistics and Bayes Factors now offer additional context to p values in the Results section, and weight of evidence is now also directly stressed in verbal descriptions throughout the Results section. This directly addresses the reviewer’s concern regarding Type II errors, as does the following addition to the Discussion section:

“An obvious limitation of this study is the relatively small sample size of N=32 in the healthcare assistants sample and N=50 in the control sample. This was due to practical limitations on access to care home facilities and their staff, and we aimed to remedy this by including a relatively large number of trials per condition in the preferential looking task (64 trials per condition per participant). In addition, where this was not possible, we have aimed to report statistics in such a way that the uncertainty in estimates is clearly reflected in both confidence intervals and Bayes Factors.” (section “Discussion”)

While on the topic of statistical power: the number of trials used in the preferential looking task is actually relatively high, and this offers relatively high statistical power because in linear mixed-effects analyses power depends on both the number of participants and the number of observations per participant. In fact, the number of observations per condition (5248 for the full sample, 2048 for the care home sample, and 3200 for the control sample) exceeds the recommended minimum for the detection of small effect sizes in related research of 1600 observations per condition. This is now discussed in the Methods section:

“For linear mixed-effects analyses, power depends on the number of individuals and the number of observations per individual; and the recommended total number of observations per condition is 1600 (Brysbaert & Stevens, 2018). In our study, the number of observations per condition is 5248 for the full sample (82 participants, 16 stimuli per condition, each with 4 repetitions), 2048 for the healthcare assistant sample, and 3200 for the control sample. Note that the cited recommendation is for research on response times, and aimed at “very small” effect sizes (Brysbaert & Stevens, 2018). By contrast, the main effect of stimulus type (disgust vs. neutral) on dwell time in preferential looking tasks is rather large: β=0.49 in a well-powered study with N=101 and 4848 observations per condition (Dalmaijer et al., 2021). In sum, while our sample size is limited, we made up for this with a relatively high number of observations per participant to meet recommendations for statistical power.

 In addition to the above, we report Bayes Factors for comparisons between linear mixed-effects models, and 95% confidence intervals on effect estimates to contextualise p values.”

(section “Statistical power”)

ISSUE 2: INTERPRETATION OF HABITUATION EFFECT

The manuscript detects a negative relation between disgust sensitivity and length of time working in health care facilities. It interprets this effect as evidence of habituation. An alternative is (at least) equally plausible: that disgust sensitivity negatively predicts exiting the field. The manuscript mentions this possibility only in the discussion. It should be mentioned more prominently earlier in the manuscript, as should the possibility of selection bias (explaining differences between health care and control groups). The design can't distinguish between these explanations for the data, and favoring one (in my mind equally-plausible) explanation except at the very end of the manuscript risks misleading readers.

ANSWER: We agree with the reviewer, and had indeed already highlighted this alternative explanation in the Discussion section. As per the reviewer’s suggestion, we have now also included it in the Results section:

“This suggested that healthcare assistants’ pathogen disgust sensitivity reduced over the duration of their careers, or that more disgust-sensitive healthcare assistants were more likely to exit the profession.” (section “Self-reported pathogen disgust”)

Note that this section used to be called “Self-reported pathogen disgust reduces over time”, but is now titled “Self-reported pathogen disgust”. While the previous version was an accurate reflection of the pattern of results, we felt this was potentially misleading as it does implicitly suggest a reduction of disgust sensitivity within individuals rather than the equally plausible reduction of disgust-sensitive individuals in the profession.

We have also further addressed the possibility of selection bias into healthcare professions by lower disgust-sensitive individuals. This was previously addressed in the Introduction and Discussion sections, and now also features in the Results section:

“The potential difference could be a result of habituation in healthcare assistants, but it could also be a pre-existing difference.” (section “Self-reported pathogen disgust”)

ISSUE 3: SOME RELEVANT LITERATURE MISSED

The manuscript would benefit from citation/discussion of at least two relevant papers.

First, Kupfer (2018) presents a compelling case that at least some of the disgust reported toward injuries is distinct from pathogen disgust. The manuscript makes a good point that assessments of "animal reminder" disgust are nearly indistinct from "core" disgust (citing Tybur et al., 2009). That (minimal) distinction might be limited to the items on the DS-R rather than pictures of dislocated joints and broken bones, though. Some commentary on this issue would be nice.

Second, Karinen et al. (2023) reports high self-other agreement in (pathogen) disgust sensitivity. The manuscript currently states, "This is an important piece of triangulation, because self-report measures and actual behavior do not always align in disgust." This sentence should be more precise (what does "always" mean, what categorizes a measures as "actual" behavior, what is the validity of such assessments, and what type of effect size does "align" refer to?). It should also direct readers to evidence for the validity of disgust sensitivity instruments, such as that reported by Karinen et al.

ANSWER: These are both excellent suggestions, and we have incorporated them into the Discussion section.

Previously, our discussion of the differences between disgust for bodily effluvia and for body envelope violations focussed on physiology:

“Previous studies have shown differences in response profiles between core and gore disgust. Bodily effluvia inspire a physiological change in stomach rhythm whereas body envelope violations alter heart-rate variability (Shenhav & Mendes, 2014). In addition, the two share a different but partially overlapping response profile in oxygenated blood in the brain (Wright et al., 2004); and in alpha power in the electroencephalogram, although this could simply reflect differences in arousal (Sarlo et al., 2005).” (section “Evidence for limited generalisation of disgust”)

We have now extended this with the suggested Kupfer reference:

“It has been argued that this difference is due to vicarious feelings inspired by empathic simulation of observed body envelope violations, which is then verbalised as “disgust” despite it being qualitatively different to disgust to bodily effluvia (Kupfer, 2018).” (section “Evidence for limited generalisation of disgust”)

In light of this, we have also changed our contextualisation of Tybur et al. (2009), which now reads:

“In self-report measures, both core and gore disgust elicitors load onto the same “pathogen disgust” factor (Tybur et al., 2009). However, this could be due to a lack of number and richness of items in the employed questionnaire, which does not include a great deal of body envelope violations.” (section “Evidence for limited generalisation of disgust”)

This is followed by the previously cited sections on how “core” and “gore” disgust are different, and in the next paragraph we now summarise the lack of behavioural avoidance to both categories as such:

“Healthcare assistants’ lack of behavioural avoidance of both bodily effluvia and bodily harm could thus reflect a genuine habituation to each separate category.” (section “Evidence for limited generalisation of disgust”)

Second, we thank the reviewer for pointing out the Karinen et al. (2023) study, and have incorporated this in our Introduction:

“The summarised studies mostly rely on self-report measures of disgust elicited by written or picture stimuli. Such self-report measures can have excellent psychometric properties, for example the internal consistency (Tybur et al., 2013) and external agreement with others’ ratings (Karinen et al., 2023) of the Three-Domain Disgust Scale, or the high test-retest reliability of self-reported disgust for images (Dalmaijer et al., 2021).” (section “Introduction”)

In the Discussion section, in the paragraph that the reviewer cited, we now return to this in the following way:

“Crucially, our findings support evidence of habituation in self-reported disgust sensitivity with corroborating behavioural data. This is an important piece of triangulation, because self-report measures and actual avoidance behaviour do not always align in disgust (Cougle et al., 2007; Dalmaijer et al., 2021; Rouel et al., 2018). For example, minor reductions in self-reported disgust (d=0.38 in two separate experiments) have been paired with a statistically significant absence of habituation in oculomotor disgust avoidance in a short-term experimental study (Dalmaijer et al., 2021), In the introduction, we suggested this could be due to demand effects. However, we think our findings in healthcare assistants suggest that long-term habituation to disgust occurs in both self-report and behavioural avoidance.” (section “Evidence for disgust habituation”)

The above addresses the reviewer’s request to be more precise in our meaning. Elsewhere, we also outline the reliability and validity of both questionnaire measures (like the evidence reported by Karinen et al.) and behavioural avoidance measures. (More on this in one of the reviewer’s other comments, specifically that on the reliability of experimental measures.)

A few other comments:

* "Disgust occurs in response to offensive stimuli." This description is circular (see Tybur et al., 2013). Stimuli are considered "offensive" because they elicit disgust. This type of definition effectively boils down to "Disgust is elicited by disgusting stimuli."

ANSWER: This is an entirely fair comment, and our definition here is merely quoting Darwin’s (cited). We have now improved the clarity of the definitional sentence by more clearly highlighting the type of stimuli (i.e. those associated with contamination and illness) and disgust’s role (i.e. avoidance).

“Disgust occurs in response to “offensive” stimuli (Darwin, 1872) in order to avoid those associated with risk of contamination and illness (Angyal, 1941).” (section “Introduction”, first paragraph)

* "A major risk here is that disgust-exposed participants simply displayed demand effects..." There's no evidence for this possibility, and the manuscript doesn't describe any reason to suggest that the possibility is plausible (or worth considering).

ANSWER: We acknowledge that we failed to cite evidence behind our thinking here. This comes from our assumption that participants are likely to perceive a demand effect in habituation-testing designs, and the fact that placebo effects are much larger for self-reported disgust than for gaze dwell-time measures of behavioural avoidance: d=1.25 versus d=0.44 (Schienle et al., 2016, International Journal of Psychophysiology).

We base our assumption of participants being able to perceive a demand effect on our notion that participants in both experimental and observational studies are quite likely to understand what the study is testing. For example, we have repeatedly asked participants for self-reported disgust ratings while repeatedly shown them the same disgusting stimuli (Dalmaijer et al., 2021, JEP:General). In observational studies like our current manuscript, workers frequently volunteer being less sensitive to disgust once they learn what our study is about.

We have now clarified this in the manuscript in two places. First, after outlining the reliability of self-report and behavioural avoidance measures, we write the following:

“However, a major difference between self-report and behavioural measures of disgust avoidance is that self-reported disgust shows a very large (Cohen’s d=1.25) placebo effect, whereas oculomotor avoidance (gaze dwell time) in preferential looking tasks only shows a small to medium (Cohen’s d=0.44) effect (Schienle et al., 2016).” (section “Introduction”)

We then return to this in the next paragraph, writing:

“Self-reported disgust shows varied levels of habituation (Cougle et al., 2007; Dalmaijer et al., 2021; Rouel et al., 2018) that can occur even in the absence of behavioural changes (Dalmaijer et al., 2021). Because of the difference in suggestibility between self-report and behavioural avoidance measures of disgust, it could be that the apparent difference in habituation between them reflects a demand effect: We think it likely that participants who are repeatedly asked to self-report disgust between being subjected to disgust-inducing stimuli are likely to understand researchers might expect them to reduce their ratings. We thus think that to better understand underlying emotional changes as a function of prolonged exposure to disgust elicitors, one must also measure behaviour.” (section “Introduction”)

As we hope the reviewer will note, where we previously wrote about “a major risk”, we now more carefully talk about what “we think” when outlining why we consider it possible for there to be a demand effect, and why we consider it important to measure behavioural avoidance in addition to self-reported disgust.

* "...one must thus measure behavior." There's a widespread assumption that behavioral measures are more valid than self-report ones. Dang et al. (2020) are (to me, persuasively) that behavioral measures are often LESS VALID than self-report ones because of their poor reliability and ambiguous validity (relative to many self-report measures). I'm not arguing that the behavioral measures adopted here suffer from these limitations. But default skepticism toward self-report measures in favor of "behavioral" ones is misguided.

ANSWER: This is a pet peeve that we share with the reviewer. Too often is it just assumed that behavioural or physiological measures are better than self-report because of assumptions of “objectivity”, whereas the reliability of such measures is often questionable. It is exactly this that has driven us to assess the psychometrics of behavioural disgust avoidance in preferential looking tasks in our past work.

We’ve demonstrated that behavioural avoidance has a test-retest reliability that is equal to self-reported disgust for the same stimuli, both with ICCs of around 0.6 when comparing 24 presentations each of two different stimuli showing faecal matter (Dalmaijer et al., 2021, JEP: General). We’ve also shown that the internal consistency of behavioural avoidance is high when measured with gaze (Spearman-Brown ρ = 0.81 to 0.92, Cronbach α = 0.78 to 0.89) and the cursor-locked aperture technique employed in the current study (Spearman-Brown ρ = 0.89 to 0.94, Cronbach α = 0.86 to 0.91) (Anwyl-Irvine et al., 2021, Behaviour Research Methods).

Furthermore, self-reported disgust sensitivity correlates with gaze avoidance (R=0.28 to 0.34) (Dalmaijer et al., JEP: General), and self-reported disgust ratings correlate with both gaze and aperture-locked mouse avoidance (R=0.19 to 0.50) (Anwyl-Irvine et al., 2022, Behavior Research Methods – note that we computed disgust approach and thus reported negative correlations in that paper, but due to the nature of the computation the sign can be flipped for disgust avoidance).

We have now clarified this in the section that the reviewer cited:

“Such self-report measures can have excellent psychometric properties, for example the internal consistency (Tybur et al., 2013) and external agreement with others’ ratings (Karinen et al., 2023) of the Three-Domain Disgust Scale, or the high test-retest reliability of self-reported disgust for images (Dalmaijer et al., 2021). While experimental tasks are often less reliable than self-report measures (Dang et al., 2020), behavioural disgust avoidance measured in preferential looking tasks has excellent internal consistency, Spearman-Brown ρ=0.81 to 0.94 and Cronbach’s α=0.78 to 0.91 (Anwyl-Irvine et al., 2022), and test-retest reliability equal to self-reported disgust ratings (Dalmaijer et al., 2021). Behavioural disgust avoidance correlates with self-reported disgust sensitivity, R=0.28 to R=0.34 (Dalmaijer et al., 2021), and with self-reported disgust ratings for stimuli, R=0.19 to 0.50 (Anwyl-Irvine et al., 2022).” (section “Introduction”)

* The manuscript describes Kendall's tau as more powerful than Pearson or Spearman r. I think that this is wrong, based on my own background in nonparametric statistics and on some sources I found online (https://www.ncss.com/wp-content/themes/ncss/pdf/Procedures/PASS/Power_Comparison_of_Correlation_Tests-Simulation.pdf). I looked at the manuscript cited in support of Kendall's being more powerful, and I don't think that it actually supports that statement. There ARE reasons to favor Kendall's over the others, but I don't think power is such a reason. Regardless, I urge the authors to carefully review this issue.

ANSWER: This one is on the senior author (sorry!), who read Bonett & Wright (2000) years ago and has evidently over-interpreted some of their tables. He has run the simulations, and confirmed the reviewer’s notion is correct: tau is indeed not more powerful than R! We now report both, and have rewritten the previously erroneous statement:

“Within the healthcare assistant sample, self-reported disgust sensitivity scores were also correlated with the number of months participants had worked in care homes, using parametric (R) and non-parametric (Kendall’s τ) correlation coefficients.” (section “Data reduction and statistical analysis”)

* In the "within group confirmatory analyses" section, the term "and an interaction" appears twice. Please clarify - interaction between which variables?

ANSWER: We have now clarified that these interactions are “between stimulus and repetition” in both instances (section “Within-group confirmatory analyses”).

That's it from me. Hope the comments are helpful. And I commend the authors on writing a fine manuscript and asking an interesting research question.

ANSWER: The comments were indeed helpful, and we are very grateful to the reviewer’s close reading and their constructive comments!

Reviewer #2

Review of PONE-D-23-30077: Long-term disgust habituation with limited generalisation in care home workers.

The manuscript reports a study that compares two groups of participants—healthcare assistants working in residential care homes and a control group. Participants do a task in which it is measured how long they look at pictures that are disgusting versus not-disgusting. Participants also self-report their disgust sensitivity. The results show that participants in the control group look more at the not-disgusting than disgusting images. This difference is not observed for the healthcare assistants. Results also show that self-reported pathogen disgust sensitivity is lower for the healthcare assistants and that it correlates with the time they are working as healthcare assistants (those having worked longer reporting lower disgust sensitivity).

Overall, I like the research reported in this manuscript, but I think some things should be improved before publication.

Positive remarks:

(1) I think the research question is important. If and how there is habituation for disgust is a puzzle. Over a decade ago, I wrote a grant proposal on this question. It did not get funded. On the one hand, habituation occurs for many stimuli, suggesting that there should be habituation for disgust-elicitors. On the other hand, a straightforward hypothesis of a functional perspective is that there should be no habituation. If the purpose of a disgust response is to reduce contact with contagious objects—then in order to fulfil that function, there should be no habituation. (Or at least, it is difficult to see how disgust could serve the function of avoiding contamination if it did habituate after repeated exposure to contaminants.)

(2) I agree with the assessment of the authors that there is currently little empirical work on the habituation of disgust. Hence, all rigorous empirical work is welcome in my opinion.

(3) I appreciate the clear mention of the limitations regarding selection bias and longitudinal selection bias. While this certainly is a limitation, I think it is better to report the results (descriptively and with acknowledging this limitation) rather than to block publication for doubts about causal inference.

ANSWER: We thank the reviewer for their positive remarks, and empathise with the highlighted difficulty in raising funds for disgust research through grant proposals.

Suggestions for improvement:

(4) In the section “Participants”, please clarify if the participants in the control condition were/were not working as healthcare assistants. The current formulation is ambiguous. If it is possible that the participants in the control group working as healthcare assistants, then I think this has substantial consequences for interpreting the results. Then the control group would no longer be a control group.

ANSWER: We have now clarified that this group was sampled from the general population, and not healthcare workers.

“The other sample was a control group sampled from the general population (not healthcare workers), [...]” (section “Participants”)

Note that individuals in the control sample could have chosen to participate regardless of their occupation (we can’t ask participants for proof of non-employment in a care home), and that Prolific does not offer enough detail on participants’ occupation in order to post-hoc exclude all disgust-facing roles. We note this limitation in the manuscript:

“We were unable to exclude participants on the basis of fine-grained occupational data, so individuals with disgust-facing roles may have been included.” (section “Participants”)

(5) In my view, this work needs a clear discussion of statistical power. The groups were relatively small (N = 32 for the group of healthcare assistants, N = 50 for control). Please report sensitivity (i.e., what effect size could likely have been detected with this sample size), for the behavioral avoidance task, but also for the results for self-reported disgust sensitivity (for comparisons between groups as well as the correlations with career time).

ANSWER: This is a very important point, and a section on statistical power is now included in the methods section. In addition, 95% confidence intervals and Bayes Factors are now included for all tests in the result section, to contextualise yes/no decisions of statistical significance on the basis of p-values.

While the reviewer is absolutely correct on the sample size being relatively small, it should be noted that this is primarily a problem for tests of differences and correlations of averages (e.g. mean dwell time within a single condition) and/or single values (e.g. duration of work in a care home or TDDS subscale score). This is now discussed in the Methods section, and power for the exact sample sizes is now shown in Figure 1:

“The sample size of this study was primarily determined by practical limitations, such as access to healthcare assistants working in care homes. We thus provide power estimations for our achieved sample sizes (N=32 healthcare assistants and N=50 control) in Figure 1. Our statistical power is low for direct tests of differences and correlations, which could result in erroneous rejection of the alternative hypothesis (Type II error). We have aimed to remedy this to some extent by contextualising p values with 95% confidence intervals on test statistics and with Bayes Factors to quantify the weight of evidence for the alternative or null hypotheses.” (section “Statistical power”)

It should be noted that the number of trials used in the preferential looking task is actually relatively high, and this offers relatively high statistical power. In fact, the number of observations per condition (5248 for the full sample, 2048 for the care home sample, and 3200 for the control sample) exceeds the recommended minimum for the detection of small effect sizes in related research of 1600 observations per condition. This is now discussed in the Methods section:

“For linear mixed-effects analyses, power depends on the number of individuals and the number of observations per individual; and the recommended total number of observations per condition is 1600 (Brysbaert & Stevens, 2018). In our study, the number of observations per condition is 5248 for the full sample (82 participants, 16 stimuli per condition, each with 4 repetitions), 2048 for the healthcare assistant sample, and 3200 for the control sample. Note that the cited recommendation is for research on response times, and aimed at “very small” effect sizes (Brysbaert & Stevens, 2018). By contrast, the main effect of stimulus type (disgust vs. neutral) on dwell time in preferential looking tasks is rather large: β=0.49 in a well-powered study with N=101 and 4848 observations per condition (Dalmaijer et al., 2021). In sum, while our sample size is limited, we made up for this with a relatively high number of observations per participant to meet recommendations for statistical power.

 In addition to the above, we report Bayes Factors for comparisons between linear mixed-effects models, and 95% confidence intervals on effect estimates to contextualise p values.”

(section “Statistical power”)

(6) In the introduction, it says “In the past, disgust avoidance behaviour has been quantified with eye-tracking measures. In preferential looking tasks, individuals look away from disgust elicitors and prefer to look at neutral stimuli instead.” Could the authors elaborate how this relates to the assumed function of pathogen disgust? They write that “you can’t reason you way around a big stinky poo.” Yes, but to avoid touching a poo, it also does not help to look away. To avoid stepping in dog poo, you must look where it is and step aside. In this context, please also clarify what is meant with adaptive/maladaptive (alluded to in the first paragraph of the introduction). I can see how in an experimental task, it is “adaptive” to look the other way (avoiding unpleasant feeling), and that for healthcare workers it might appear “maladaptive” to feel so much disgust during their work. However, this usage of adaptive and maladaptive are different from the meaning of these terms in a functional evolutionary perspective: Disgust is likely the output of some adaptation—probably an adaptation for pathogen avoidance, so one could say that it is adaptive to feel disgust towards pathogen hazards, but maladaptive to feel disgust towards objects that are safe.

ANSWER: The reviewer makes several good points here, and we have tried to address them all.

The reviewer asked how gaze avoidance behaviour maps onto avoidance behaviour in the outside world, where gaze is necessary to find offensive stimuli. We now outline how gaze avoidance in this task is quite stereotyped, and typically shows a brief period of initial disgust approach. This is analogous to encountering a “big stinky poo” in the wild: first we observe that it’s there, and then we can avoid it after learning its location.

“Sustained avoidance in this task is typically preceded by a brief (~1 second) period of initial approach to novel disgusting stimuli (Armstrong et al., 2022; Dalmaijer et al., 2021; Nord et al., 2021).” (section “Introduction”)

It might also be reassuring to know that the task has external validity: it correlates decently with DS-r scores across two different experiments in some of our previous work: R=0.28 in N=104 and R=0.34 in N=99 (Dalmaijer et al., 2021, JEP:General).

The reviewer also asked for clarification on how avoidance in a preferential looking We have now elaborated on how a preferential looking task reflects the function of pathogen disgust, and specifically what we mean by “adaptive” within the task, and how this maps onto adaptive behaviour outside of the task. We have now made this explicit in the introduction:

“The task is therefore analogous to coming into contact with offensive stimuli in the wild: after perceiving a stimulus is offensive, the feeling of disgust drives avoidance of potential contaminants and is adaptive because it reduces the risk of illness.” (section “Introduction”)

Finally, the reviewer made an important point about whether disgust avoidance is always adaptive or maladaptive. Specifically, they highlight an important nuance: avoidance is adaptive for pathogen hazards, but maladaptive towards non-hazards. In practice, healthcare assistants have to deal with pathogen hazards whether they are disgusted by them or not, and it’s their hygiene behaviour that actually allows them to avoid potential consequences. In this light, disgust would only be adaptive if it improved adherence to hygiene behaviour. We now clarify this in the first paragraph of the introduction:

“While disgust is evolutionary selected for at the population level in a wide variety of environments (Dalmaijer & Armstrong, 2020), disgust sensitivity is not necessarily related to current health within individuals (de Barra et al., 2014). Hence, provided hygiene is observed, feelings of disgust could be maladaptive in certain modern contexts.” (section “Introduction”)

(7) The authors mention as a possible practical implications that job satisfaction among healthcare workers might be increased with pharmacological support that facilitates habituation of disgust responses. I am not sure this is a good idea. In my view, it would be good to mention the possibility that this has perverse (adverse) consequences. For example, suppose that some drugs makes these healthcare workers have such dampened disgust responses that they do not wash their hands after touching a big stinky poo. Or start having fun by throwing poo at each other. Is that “adaptive”?

ANSWER: The reviewer is right to be cautious, and we have adjusted this section accordingly. Our reasoning was that domperidone has a rather subtle effect, and we thus didn’t anticipate any effects in the “wrong” direction (i.e. being too lax due to reduced disgust). The following text was added to the manuscript:

“However, further research is warranted before implementation of this type of measure: while disgust is unpleasant, it is also adaptive. The effect of a single dose of domperidone on disgust avoidance was subtle, but it is unknown if sustained usage could push individuals towards maladaptive behaviour such as reduced hand-washing.” (section “Potential implications for healthcare staff and retention”)

---

## [Decision Letter · Decision Letter 1]

12 Feb 2024

Long-term disgust habituation with limited generalisation in care home workers

PONE-D-23-30077R1

Dear Dr. Dalmaijer,

We’re pleased to inform you that your manuscript has been judged scientifically suitable for publication and will be formally accepted for publication once it meets all outstanding technical requirements.

There is only one important point raised by R2 for the introduction of the paper. I ask that you make the necessary revision before submitting the final version of the paper.

Within one week, you’ll receive an email detailing the required amendments. When these have been addressed, you’ll receive a formal acceptance letter, and your manuscript will be scheduled for publication.

Kind regards,

Cengiz Erisen

Academic Editor

PLOS ONE

Additional Editor Comments (optional):

Reviewers' comments:

Reviewer's Responses to Questions

**Comments to the Author**

1. If the authors have adequately addressed your comments raised in a previous round of review and you feel that this manuscript is now acceptable for publication, you may indicate that here to bypass the “Comments to the Author” section, enter your conflict of interest statement in the “Confidential to Editor” section, and submit your "Accept" recommendation.

Reviewer #1: All comments have been addressed

Reviewer #2: All comments have been addressed

2. Is the manuscript technically sound, and do the data support the conclusions?

Reviewer #1: Yes

Reviewer #2: Yes

3. Has the statistical analysis been performed appropriately and rigorously? 

Reviewer #1: Yes

Reviewer #2: Yes

4. Have the authors made all data underlying the findings in their manuscript fully available?

Reviewer #1: Yes

Reviewer #2: Yes

5. Is the manuscript presented in an intelligible fashion and written in standard English?

Reviewer #1: Yes

Reviewer #2: Yes

6. Review Comments to the Author

Reviewer #1: The authors did a very nice job of responding to my comments. The study is not perfect, but they have done a great job of describing and contextualizing its limitations. Even with these limitations, it makes a nice contribution to the literature.

Reviewer #2: In my view, the authors have addressed the comments from the reviewers. Therefore, I recommend publication. In my view, there is a minor issue that could be corrected (this is not worth a "revision" in my view), see below.

Minor issue:

The opening paragraph now includes the following: "While disgust is evolutionary selected for at the population level in a wide variety of environments (Dalmaijer & Armstrong, 2020), disgust sensitivity is not necessarily related to current health within modern humans (de Barra et al., 2014). Hence, provided sufficient hygiene is observed, feelings of disgust could be maladaptive in certain contexts."

I think here the revisions did not improve the text. "While" indicates a contradiction. But there is none. The correlation (or absence of it) between health and disgust sensitivity in modern contexts is independent from selection that shaped the emotion in the past environment. "Hence" means that it logically follows. It does not. The conclusion that follows can be true regardless of the stated premises. It seems to me that the authors argue to reach a claim (provided sufficient hygiene is observed, feelings of disgust could be maladaptive in certain contexts). I agree with this claim. But it should not be presented as the logical conclusion from the above premises. I think it does not follow. I think it would be better to leave the incorrect argument out.

For example: "It [disgust] is counted among the basic emotions (Ekman & Friesen, 1971), and thought to be evolved in response to pathogen threat (Curtis & Biran, 2001; Rozin & Fallon, 1987; Tybur et al., 2013). However, provided sufficient hygiene is observed, feelings of disgust could be unpleasant or counter-productive in certain contexts."

7. PLOS authors have the option to publish the peer review history of their article (what does this mean?). If published, this will include your full peer review and any attached files.

Reviewer #1: **Yes: **Josh Tybur

Reviewer #2: No

---

## [Editor Report · Acceptance letter]

22 Mar 2024

PONE-D-23-30077R1 

PLOS ONE

Dear Dr. Dalmaijer, 

I'm pleased to inform you that your manuscript has been deemed suitable for publication in PLOS ONE. Congratulations! Your manuscript is now being handed over to our production team.

Kind regards, 

on behalf of

Dr. Cengiz Erisen 

Academic Editor

PLOS ONE